# Heavy Metal-Associated Isoprenylated Plant Proteins (HIPPs) at Plasmodesmata: Exploring the Link between Localization and Function

**DOI:** 10.3390/plants12163015

**Published:** 2023-08-21

**Authors:** Zoe Kathleen Barr, Tomáš Werner, Jens Tilsner

**Affiliations:** 1Biomedical Sciences Research Complex, University of St Andrews, BMS Building, North Haugh, St Andrews, Fife KY16 9ST, UK; zkb1@st-andrews.ac.uk; 2Cell & Molecular Sciences, The James Hutton Institute, Dundee DD2 5DA, UK; 3Department of Biology, University of Graz, Schubertstraße 51, 8010 Graz, Austria

**Keywords:** plasmodesmata, heavy metal-associated plant proteins, HIPP, prenylation, metallochaperone, abiotic and biotic stress, cytokinin

## Abstract

Heavy metal-associated isoprenylated plant proteins (HIPPs) are a metallochaperone-like protein family comprising a combination of structural features unique to vascular plants. HIPPs possess both one or two heavy metal-binding domains and an isoprenylation site, facilitating a posttranslational protein lipid modification. Recent work has characterized individual HIPPs across numerous different species and provided evidence for varied functionalities. Interestingly, a significant number of HIPPs have been identified in proteomes of plasmodesmata (PD)—nanochannels mediating symplastic connectivity within plant tissues that play pivotal roles in intercellular communication during plant development as well as responses to biotic and abiotic stress. As characterized functions of many HIPPs are linked to stress responses, plasmodesmal HIPP proteins are potentially interesting candidate components of signaling events at or for the regulation of PD. Here, we review what is known about PD-localized HIPP proteins specifically, and how the structure and function of HIPPs more generally could link to known properties and regulation of PD.

## 1. Introduction

Heavy metal-associated isoprenylated plant proteins (HIPPs) make up one of two families of plant metallochaperone-like proteins containing a heavy metal-associated (HMA) domain [1,2,3]. HIPPs and their related group, heavy metal-associated plant proteins (HPPs), are defined by possessing one or two HMA domains (Figure 1A). The HMA domain contains two conserved cysteine residues in a CXXC motif (where ‘X’ is any amino acid), which function in complexing heavy metal ions. The HMA domain is a conserved, autonomously folding domain consisting of approximately 70 amino acids and adopting a ferredoxin-like βαββαβ sandwich fold (PFAM: PF00403, InterPro: IPR006121). This characteristic fold was confirmed in the HIPP family by the crystal structure of the rice (*Oryza sativa* L.) protein OsHIPP19 [4] (PDB: 7B1I), Figure 1B). Furthermore, AlphaFold structure predictions [5,6] of Arabidopsis (*Arabidopsis thaliana* (L.) Heynh.) HIPPs consistently predict the HMA fold with high or very high confidence (pLDDT > 70) while much of the remaining structure is often assigned low confidence and includes regions of predicted intrinsic disorder [7].

In addition, HIPPs, but not HPPs, are further defined by an isoprenylation site consisting of a C-terminal CaaX motif (where ‘a’ is an aliphatic residue and ‘X’ is any amino acid). Protein prenyl transferases covalently link the cysteine in this motif to a hydrophobic isoprenyl farnesyl (C15) or geranylgeranyl (C20) chain via a thioether bond. The C-terminal -aaX residues are subsequently proteolytically cleaved, leaving the isoprenylated cysteine, whose carboxyl group becomes methylated, as the new protein C terminus [8]. Protein prenylation is important to mediate protein–protein interactions as well as to act as a membrane anchor either by direct intercalation of the prenyl group with phospholipid membranes or through binding to membrane-associated prenyl-binding proteins [9,10,11]. The combination of the HMA domain and the isoprenylation site that characterizes the HIPP protein family has uniquely evolved only in vascular plants [2]. 

The HIPP family is diverse, with nearly 50 members in Arabidopsis [1,2,12]. HIPPs have been implicated in mediating plant tolerance to heavy metals, as well as impacting plant development, growth, and stress responses [3]. Similarly, protein isoprenylation is often linked to stress responses in plants [8]. Thus, HIPPs appear to be important, likely membrane-associated factors in plant metal homeostasis and stress responses, but the majority of these proteins have so far not been functionally characterized. Recently, a substantial subset of HIPPs—around one-fifth of Arabidopsis isoforms—have been found in plasmodesmal proteomes, with localization to plasmodesmata confirmed by live-cell imaging in some cases (Table 1, Figure 1C,D).

Plasmodesmata (PD), the cell junctions of plants, are membrane-lined nanochannels spanning the cell wall that provide cytoplasmic, endoplasmic reticulum (ER), and plasma membrane (PM) continuity between neighboring cells. PD-mediated connectivity supports nutrient distribution, developmental signaling, biotic and abiotic stress responses, phloem loading, and systemic signaling, and is essential for terrestrial plant life (for review: [13,14,15,16]). This connectivity can in turn be exploited by pathogens, including viruses that transport their genomes cell-to-cell through PD, as well as bacterial and eukaryotic pathogens, which modify PD function to interfere with defense signaling or possibly to prepare naïve cells for infection [17,18,19]. 

PD are membrane-rich structures, comprising a PM-lined channel inside of which an ER tubule is also continuous between cells [14]. Transport through PD occurs mainly through the cytoplasmic compartment between the plasmodesmal PM and ER, known as the cytoplasmic sleeve. The molecular size exclusion limit of PD is actively regulated, with the most well-characterized mechanism being the accumulation and removal of callose (β-1,3-glucan) in the cell wall region around PD openings [20,21]. An increase in callose in this area is thought to push the PM inward, decreasing the cross-section of the cytoplasmic sleeve. Apart from regulating intercellular transport, PD are also signaling hubs where distinct membrane-based signaling pathways operate [22,23]. Many aspects of PD-localized signaling, the regulation of callose turnover, and how these processes intersect at PD are still only incompletely understood.

Therefore, it is of vital interest to identify and functionally characterize proteins that are targeted to PD. The recently discovered PD localization of metallochaperone-like HIPP proteins with potential functions in metal homeostasis, stress responses, and plant development suggests that they might play roles in the regulation of PD-localized molecular processes. However, so far, no PD-specific functions of these putative plasmodesmal HIPPs have been characterized. Here, we bring together knowledge of HIPP functions in general, including PD-localized isoforms specifically, and consider where these intersect with PD involvement in stress responses and development to develop a conceptual framework for characterizing HIPP functions at PD.

## 2. HIPP Proteins Identified in PD Proteomic Studies

There have been several successful attempts to define PD proteomes. Each of these was produced by alternative methods ranging from plasmodesmata purification to meta-analysis-based in silico prediction and sources included suspension cell cultures [24,25] and 12-week-old plants [26] of Arabidopsis, naïve or Turnip mosaic virus-infected leaves of *Nicotiana benthamiana* Domin [27], suspension cell cultures of *Populus trichocarpa* Torr. & A.Gray ex. Hook. [28], and the moss *Physcomitrium patens* (Hedw.) Bruch & Schimp. [29,30]. Most of these PD proteomes contain HIPPs (Table 1), as well as a single HPP (HPP01). HPPs are a smaller family, with approximately half as many members as HIPPs in Arabidopsis [1] so whilst further proteomic advances may later change this view, the lower abundance of HPPs in PD proteomes is suggestive that the addition of isoprenylation in HIPPs may be key for PD association of this protein family. For three of these putative PD HIPPs, at least partial PD localization has been confirmed experimentally (Figure 1C,D, Table 1).

**Figure 1 plants-12-03015-f001:**
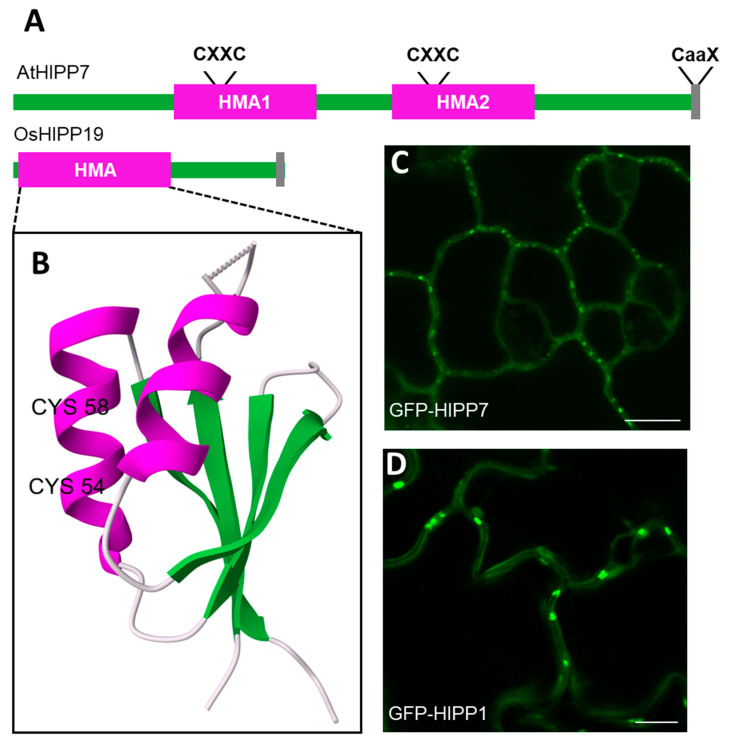
HIPP proteins and their PD association. (**A**) Schematic of the domain structure with one (OsHIPP19: Q01IL6) or two (AtHIPP7: Q9C5D3) HMA domains. The HMA domain (magenta) and the prenylation site (grey) are annotated with their respective motifs. (**B**) Crystal structure of OsHIPP19 HMA domain. Cysteine residues of the CXXC motif are labeled and the characteristic βαββαβ fold is shown with beta strands (green) and alpha helices (magenta). Image from the RCSB PDB (RCSB.org) of PDB ID 7B1I [4,31,32]. (**C**) Confocal microscopy of Arabidopsis leaf epidermal cells stably expressing UBQ10:GFP-HIPP7 (**C**) and UBQ10:GFP-HIPP1 (**D**). Scale bars, 10 µm. Adapted with permission from Ref. [33]. Copyright 2021, Cell Press (Elsevier).

**Table 1 plants-12-03015-t001:** PD-associated HIPP and HPP proteins. The supplementary data tables of available PD proteomes [24,25,26,27,28,29,30] were searched for HIPP and HPP-specifying terms (“*HIP*”, “*HPP*”, “*HMP*”, “*HMA*”, “*metal*”, “*FP*”, and “*isoprenyl*”). For each identified protein, the closest Arabidopsis homolog was aligned by accession number with the listed HIPPs and HPPs compiled from previous HIPP studies [1,2,12,34].

TAIR ID	Tehseen et al. [1]/de Abreu-Neto et al. [2] nomeclatures	Dykema et al. [34] nomenclature	Li et al. [12] Nomenclature	Number of HMA Domains	Clade [1]	Clade [2]	PD Proteomes/PD-Related Papers	PD-Localisation Confirmed	Known regulation, Functions and Interactions
**AT1G01490**	HIPP39		HMP01	1	VI	V	*N. benthamiana* [27]		Fruit-specific [12].
**AT2G28090**	HIPP01		HMP17	2	I	I	Arabidopsis [33]	**yes** [33]	Low expression in all tissues [12], expressed during meiosis [35]. Alters cytokinin responses and development, regulates the CKX ERAD [33].Interacts with different cytokinin-degrading CKX proteins [33].
**AT2G36950**	HIPP05	FP2	HMP20	2	I	I	Arabidopsis [25,26]		Weakly Cu-binding [36]. ↑ Cu/Cd/Zn [12], ↑ cold [37], ↑ hypoxia [38],↓ brassinosteroids [39], ↓ NO [40], ↓ cytokinin [33]. Alters cytokinin responses, regulates the CKX ERAD and ER stress [33]. Interacts with different cytokinin-degrading CKX proteins [33].
**AT3G05220**	HIPP32/HPP34		HMP23	1	III	III	Arabidopsis [24,25,26]		↑Cu/Pb/Cd/Zn [12], ↑ cold [37], ↑ NO [40], ↓ Geminivirus infection [41]
**AT3G06130**	HIPP34/HIPP32		HMP25	1	III	III	Arabidopsis [25], *N. benthamiana* [27]		↑ Zn/Pb/Cu [12], ↓ NO [40], ↓ Geminivirus infection [41]. Interacts with cytokinin-degrading protein CKX1 [33].
**AT4G16380**		FP1	HMP35	1 *			Arabidopsis [24,25,26], *P. trichocarpa* [28]		↑ Cu/Cd/Zn/Pb [12]
**AT4G35060**	HIPP25		HMP39	1	IV	II	Arabidopsis [24,25]		Expressed in leaves [12], main roots, SAM, flower buds, trichomes [1], ↑ in trichomes [42]; expressed in endosperm but not embryo during germination [43]. ↓ auxin [39], ↓ NO [40], ↑ PtdInsP’s [44], initially ↑ then ↓ cold. No interaction with transcription factor ATHB29 [45].
**AT4G38580**	HIPP26	FP6	HMP40	1	IV	II	Arabidopsis [24,25], *N. benthamiana* [27,46], *P. patens* [29]	**yes** [46] ******	Binds Cd/Cu/Pb [47]. Expressed in lateral root tips, SAM, weakly in leaf vasculature [1], mostly vascular expression [45,46], ↑ in trichomes [42]. ↑ Cd [47], ↑ cold [45,48,49], ↑drought [45,46,49], ↑ salt [45], ↓ heat [50], ↑ Pomovirus infection [46]. Interacts with acyl-CoA binding protein ACBP2 [47], zinc finger homeobox domain transcription factor ATHB29 via HMA cysteines [45], metallocarboxipeptidase inhibitor TCMP-1 [51], viral movement protein [46]. Overexpression increases Cd tolerance [47], rescues Cd-sensitive yeast mutant [1]. S-acylated [46]
**AT5G03380**	HIPP06		HMP43	2	I	I	Arabidopsis [25], *P. trichocarpa* [28]		Mainly root-expressed [12]. ↑ hypoxia [52], ↑ Geminivirus infection [41], ↑ wound responses [53], ↓ cytokinin [33]. Alters cytokinin responses and development, regulates the CKX ERAD and ER stress [33]. Interacts with different cytokinin-degrading CKX proteins [33].
**AT5G19090**	HIPP33		HMP46	1	III	III	Arabidopsis [24,25], *N. benthamiana* [27], *P. trichocarpa* [28]		↑ Cu/Cd/Zn/Pb [12], ↓ NO [40], ↓ Geminivirus infection [41]. Non-canonical intron/likely alternatively spliced [54].
**AT5G50740**	HPP01		HMP52	2	I		*N. benthamiana* [27]		↑ cytokinin [55], ↓ Geminivirus infection [41]
**AT5G63530**	HIPP07	FP3	HMP54	2	I	I	Arabidopsis [25,26,33]	**yes** [33]	High general expression except in roots [12]. ↓ Geminivirus infection (Ascencio-Ibanez), ↓ cytokinin [33]. Alters cytokinin responses and development, regulates the CKX ERAD and ER stress [33]. Interacts with different cytokinin-degrading CKX proteins [33].

↑: upregulation in response to; ↓: down-regulation in response to; * 1 and 2 omit this isoform stating that pfam and NCBI both fail to predict the presence of HMA domains. However, the HMA domain is predicted by [12] and annotated in TAIR (https://www.arabidopsis.org) as predicted by INTERPRO. ** *N. benthamiana* isoform.

## 3. HIPP Localization to PD and Targeting Mechanisms

Despite recent progress in isolating PD, many potential PD components remain uncharacterized. The confirmation of protein localization, subcellular trafficking, and targeting mechanisms, functional interactions, and the structural contribution of PD candidate proteins will bring progress to understanding the wider functionality of PD. Only a small number of protein families are considered to have confirmed localization and function at PD. These have been well-summarized [56]. The HIPP family would be a significant addition to this list.

So far, PD localization has been directly confirmed by live-cell imaging for Arabidopsis HIPP1 and HIPP7 [33] and *N. benthamiana* HIPP26 [46]. It should be noted that in addition to their significant enrichment at PD, the HIPP proteins studied experimentally also display other subcellular localizations, such as at PM, the ER membrane, and in the nucleus. Interestingly, the Arabidopsis HIPP7 protein localized to PD when expressed in both Arabidopsis and *N. benthamiana*, suggesting that similar targeting mechanisms may operate in different plant species. These studies found that isoprenylation was involved in PD targeting—both HIPP7 and HIPP26 largely lost their PD localization during transient expression in *N. benthamiana* when the prenyl-accepting cysteine was mutated. On the other hand, the non-prenylated HIPP7 retained residual PD localization when stably expressed in Arabidopsis, indicating that additional targeting mechanisms exist.

It is worth noting that, in addition to isoprenylation, *N. benthamiana* HIPP26 was found to undergo another type of lipid modification by S-acylation (also called palmitoylation). The loss of this second lipidation further reduced the localization of HIPP26 to PM and PD, suggesting that both lipidations have a synergistic effect on the protein targeting [46]. Given that S-acylation generally provides stronger membrane association than prenylation [57], it will be important to investigate whether S-acylation is a common regulatory feature of other HIPPs, in particular with regard to their targeting to PD. Amongst ~600 Arabidopsis proteins identified as being S-acylated, 3 HIPPs were identified (AT1G29000.1: HIPP04, AT4G35060.1: HIPP25, and AT4G38580.1: HIPP26) [58] of which the latter two are present in PD proteomes (Table 1).

The mechanisms of protein targeting to PD are still largely unknown. There is no consensus sequence or common mechanism yet known for the localization of natively expressed and pathogenic PD proteins [56,59]. The single transmembrane domain of plasmodesmata localized protein 1 (PDLP1) is sufficient for PD targeting [60]. More recent work found a novel targeting sequence common to PDLPs and some receptor kinases [59]. An investigation of PDLP5 with machine learning techniques and molecular experiments found an unconventional targeting signal located near the membrane in the apoplastic region of PDLPs and some receptor kinases. On the other hand, apoplastic plasmodesmata callose binding protein 1 (PDCB1) and PD-associated beta-1,3-glucanase (BG_pap) require a post-translational modification, a glycosyl phosphatidyl inositol (GPI) anchor, which inserts into the outer leaflet of the PM, for PD localization. The GPI anchor also acted as a PD targeting signal for an apoplast-targeted fluorescent protein [61]. Lastly, the N-terminal region of the Tobacco mosaic virus movement protein mediates PD targeting, likely through interaction with the ER-PM contact site membrane tethering protein synaptotagmin A [62,63,64]. Thus, PD targeting mechanisms are divergent and tend to be atypical, and therefore, difficult to elucidate, but are generally linked to membranes, which is not unexpected for PD as membrane-rich structures. So far, HIPPs are the only proteins for which membrane-anchoring by isoprenylation or S-acylation has been shown to be involved in PD targeting, so it remains to be elucidated if there exists any common targeting mechanism with other PD-localized proteins.

Interestingly, as mentioned above, *N. benthamiana* HIPP26 appears to have a dual localization at PM/PD but also in the nucleus, where, similar to its Arabidopsis counterpart, it regulates the expression of drought-responsive genes [45,46]. Therefore, a change in localization from PD towards the nucleus through a reduction in membrane anchoring could play a regulatory role. Some receptor kinases have been shown to localize transiently to PD in response to stress and this requires protein phosphorylation [65]. It is therefore possible that the involvement of membrane-anchoring post-translational modifications—and particularly S-acylation, which is reversible [58]—may similarly be important for regulated or transient targeting of HIPPs during signaling processes. On the other hand, one of the HMA-containing proteins identified in PD proteomes is the non-isoprenylated HPP1 (Table 1). It would be interesting to see if this protein is indeed PD-localized, and whether no other post-translational modifications are involved in that case.

The confirmed PD localization of some HIPPs and the identification of several others in PD proteomes suggests that these proteins have functions requiring at least partial or transient PD targeting. Since, so far, no PD-specific HIPP functions have been characterized, we continue by exploring the characterized functions of HIPPs and how these may relate to PD-associated processes.

## 4. HIPPs’ Role as Metallochaperones

Early direct evidence of HIPPs metal binding activity came from in vitro study of heterologously expressed HIPP7 (FP3) interactions with metal-chelating columns showing the strongest affinity for Cu^2+^, Ni^2+^, and Zn^2+^ [34]. Suzuki et al. [66] have shown that HIPP6 (CdI19) binds Cd^2+^, Hg^2+^, and Cu^2+^ ions via CXXC of the HMA domain. In vitro-translated HIPP26-bound Cd^2+^, Cu^2+^, and Pb^2+^ in metal affinity chromatography [47]. More recently, investigations of HIPP3 used inductively coupled plasma mass spectrometry to analyze metals bound by bacterially expressed protein, and significant binding was observed only for Zn^2+^ [67]. The zinc-binding capacity was observed with the native HIPP3 but not a modified HIPP3 with mutated HMA domains. However, given the recent increased interest in the functional characterization of HIPPs, relatively few studies include a metal binding assay, and none in planta, though Dykema et al. [34] found unidentified, radioactively isoprenylated proteins binding to a copper-chelate column in extracts from tobacco BY-2 cells.

The metal-binding capacity of HIPPs through the HMA domain, together with the fact that this conserved domain occurs in several well-characterized metallochaperone proteins, such as, for example, ATX1 [68], led to the hypothesis that HIPPs may have metallochaperone functions in plants [66] and the classification of HIPPs as metallochaperone-like proteins has become the default in the literature. Metallochaperones are a distinct class of proteins that mediate the delivery of metal ions to other proteins through protein–protein interactions. This molecular activity is relevant to facilitate the function of many metalloproteins (25–50% of the total proteome are expected to bind a metal ion [69,70]) and to scavenge metal ions for detoxification [3]. Although several HIPPs can clearly bind metal ions, their metallochaperone-like activity sensu stricto (transfer of metal ions to other proteins) needs to be verified experimentally. Interestingly, the HMA domain cysteines are required for HIPP26 interaction with a zinc finger homeodomain transcription factor suggesting a regulation through a metal delivery [45]. However, it is also possible that heavy metal coordination through the HMA domain may additionally or alternatively serve other functions than chaperone activity, for instance, the homodimerization of HIPP7 [33].

On the other hand, as discussed below, many of the characterized HIPP functions are related to other abiotic stresses such as drought and cold, as well as pathogen responses, and in some cases, are mediated by HIPP protein–protein interactions for which the metal binding activity is not required [33]. In addition, common across the HIPP family are proline-rich motifs located between the HMA domain(s) and isoprenylation site [2]. These motifs form a helix, commonly referred to as ‘sticking arms’, which mediate protein–protein interactions [2] and, for instance, are bound by Src homology-3 (SH3) domains [71,72]. Deletion of a proline-rich region in OsHIPP05 (Pi21) results in functional inhibition, which in turn increases resistance to the rice blast fungus, *Magnaporthe oryzae* (T.T. Hebert) M.E. Barr [73]. Furthermore, blocking the isoprenylation of wheat HIPP1 reduced its localization to the PM and caused a loss of its function in resistance to powdery mildew (*Blumeria graminis f.sp. tritici* (DC.) Speer) [74]. Therefore, heavy metal-independent protein–protein interactions may be an important part of HIPP functions, and an exclusive focus on heavy metal stress likely limits understanding of the putative functions of HIPPs at PD. Nevertheless, it is worthwhile to summarize known interactions of PD with heavy metals.

## 5. Heavy Metal Effects on PD

A link between PD regulation and heavy metal ions has been previously established. A change to PD permeability was observed in response to treatment with excess iron, zinc, copper, or cadmium, but also in response to iron or zinc deficiency [75]. Interestingly, these responses differed between the different metals and also between tissues. Both the absence and excess of iron, as well as cadmium stress, caused a decrease in cell–cell connectivity for both root-to-shoot movements of the small symplastic tracer carboxyfluorescein (CF, 0.4 kDa) and phloem-to-meristem movement of the green fluorescent protein (GFP, 27 kDa) in Arabidopsis roots. Conversely, the absence or excess of zinc, as well as copper stress, resulted in increased connectivity for root-to-shoot diffusion of CF, whereas phloem-to-meristem diffusion of GFP was decreased in response to the lack of zinc but unaffected by excess zinc or copper. While the PD permeability-associated effects of excess iron and copper were linked to corresponding opposite changes in callose levels (i.e., increased in response to excess iron but decreased in response to copper), responses to zinc and cadmium were not significantly associated with callose metabolism. These results suggest that PD respond to various heavy metal levels in differentiated ways and possibly using both callose-dependent and -independent mechanisms. Whilst the study identified callose synthases and callose-degrading β-1,3-glucanases as mediators of the callose-dependent regulation, the metal-sensing components of these signaling pathways remain to be identified [75]. Interestingly, callose-mediated responses to heavy metal stress reduced root growth [75], and this effect was similar to that of HIPPs’ overexpression or higher-order *hipp* loss-of-function mutations in the absence of heavy metal stress [33].

## 6. HIPP Functions in Responses to Abiotic and Biotic Stresses

When considering the functions of HIPPs, it is apparent from the literature that some of their roles are indeed related to metal detoxification as might be expected from metallochaperones [3]. For instance, the overexpression of HIPP26, the most well-characterized PD-associated HIPP protein (Table 1), increases the cadmium tolerance of transgenic Arabidopsis, and HIPP26 was also shown to directly bind Cd^2+^ and its transcription is induced by cadmium stress [47]. The role of HIPP26 in cadmium detoxification is further confirmed by its ability, alongside related HIPPs 20, 22, and 27, to rescue a cadmium-sensitive yeast mutant [1]. HIPP metal detoxification functionality in rice (*Oryza sativa*) has also been established [76,77,78]. Using Cd stress as an example, OsHIPP16 overexpression improved Cd tolerance while *oshipp16* knockout mutants were found with increased Cd concentration in leaves [77]. OsHIPP19 was shown to have binding capacity for Cd and copper and moderate Cd retention at nodes and Cu uptake [78]. A further role is the HIPP-mediated accumulation of metal ions in the cell wall. Genetic experiments in maize (*Zea mays* L.) indicated that ZmHIPP facilitated lead deposition in the cell wall, thus reducing its cellular toxicity [79]. It has also been speculated that Cd accumulation in the cell wall of sugar beet (*Beta vulgaris* L.) roots may be linked to *BvHIPP32* upregulation by silicon [80].

The role of *HIPP* genes in heavy-metal homeostasis is further supported by numerous studies showing that different *HIPP* genes are transcriptionally regulated by various metal ions. For example, in Tartary buckwheat (*Fagopyrum tataricum* (L.) Gaertn.), transcriptome analysis found cadmium stress associated with 15 HMA genes including 5 of the 13 *HIPP*s identified in the species [81]. In sugar beet, the examination of *HIPP*s found *BvHIPP24* differentially expressed in Cd stress, suggesting moderate concentrations of Cd to induce expression while severe Cd stress could result in inhibition [82]. Some more examples of *HIPP*s were identified as differentially expressed genes (DEGs) during metal stress of lead in *Medicago sativa* L. (alfalfa) [83], antimony in *Boehmeria nivea* (L.) Gaudich. (ramie) [84], Cd, Cu, and manganese in rice [76], and Cu^2+^, Cd^2+^, Zn^2+^, and Pb^2+^ in Arabidopsis [12]. These include *HIPP39* [84], *HIPP05*, *HIPP34*, *HIPP32*, *HIPP33,* and *FP1* [12], whose gene products are associated with PD (Table 1).

However, other characterized functions relate to other abiotic stresses or pathogen responses. As previously mentioned, HIPP26 upregulates drought responses in Arabidopsis and *N. benthamiana* [45,46]. Various other *HIPP*s are upregulated in response to drought, cold, or hypoxia [2,37,38,48,49,85], or downregulated in response to heat stress [50] although direct evidence of a role in stress responses is still lacking in most cases. *HIPP6* (in PD proteomes, Table 1) is upregulated by wound responses [53].

Furthermore, HIPPs are also linked to biotic stress responses, e.g., *HIPP3* is upregulated in response to bacterial infection and in turn regulates the expression of large numbers of genes predominantly affecting salicylic acid signaling [67]. Several *HIPP*s are differentially regulated in response to Geminivirus infection [41] or nitric oxide [40] and *N. benthamiana HIPP26* was upregulated during Potato mop-top virus (PMTV) infection [46]. Recent studies also provided insight into the possible molecular functions of HIPP proteins during biotic interactions. For example, Cowan et al. [46] showed that HIPP26 interacts with the PMTV movement protein TGB1, which is relevant for the virus’ long-distance movement. The work also proposed that this interaction regulates HIPP26 lipidation and targeting to the nucleus, which promotes drought tolerance by activating respective transcriptional responses. In rice, the effector protein AVR-Pik of rice blast fungus interacts with the HMA of OsHIPP19 with high affinity [4] and the *Rhizoctonia solani* J.G. Kühn secreted protein RsMf8HN interacts with OsHIPP28 and acts as a trigger for host immune response [86], indicating that some HIPPs may function as susceptibility factors or/and in pathogen sensing.

## 7. Functions of HIPPs in Regulating Plant Development

PD-mediated intercellular trafficking of signaling molecules and regulatory proteins has been implicated in diverse cellular and physiological processes that are fundamental for plant growth and development [87,88,89]. Several recent studies have suggested that in addition to their function in metal homeostasis and a/biotic stress responses, HIPP proteins also play an important role in regulating plant development. Genetic experiments [33] have revealed that Arabidopsis HIPPs from the phylogenetically distinct clade I, including PD-localized HIPP1 and HIPP7, regulate multiple developmental processes in Arabidopsis, such as root meristem formation and leaf growth. The developmental changes were apparent only in higher-order *hipp* mutants, indicating a high degree of functional redundancy among the analyzed *HIPP* genes. This genetic redundancy may hamper future efforts to uncover novel functions of *HIPP* genes. The work [33] also provided insight into the molecular mechanisms underlying the activity of the studied HIPPs. The proteins have been shown to be components of the ER-associated protein degradation (ERAD) pathway, which functions in relieving ER stress, but likely also in controlling levels of specific regulatory proteins. HIPPs control the ERAD-mediated degradation of cytokinin-degrading CKX proteins [90] and thereby control cytokinin responses and respective developmental processes [33]. Interestingly, the *HIPP* genes are themselves transcriptionally controlled by cytokinin, further supporting their role in cytokinin homeostasis [33]. How this regulatory mechanism is functionally linked to PD biology is currently unclear, though there is evidence suggesting cytokinins regulate PD maturation and PD transport [91,92]. Given that the ER is closely associated with PD, together with the fact that PD are an important hub for different molecular pathways, it is tempting to speculate that specific steps of the ERAD pathway, such as the transport of ERAD client proteins across the ER membrane, may be specifically located or at least enriched at PD. This notion is supported by recent studies showing that several different ER-associated proteins involved in protein folding and quality control are targeted to PD [93,94,95,96,97]. Moreover, one of the Arabidopsis CDC48 proteins functioning in the retrotranslocation of proteins from the ER during ERAD has recently been shown to facilitate PD-mediated trafficking of the SHORT-ROOT transcription factor in Arabidopsis roots [98].

Further support for the role of HIPP proteins in plant development came from the recent study by Zhang et al. [99] showing that the *TaAIRP2-1B* gene encoding a RING finger E3 ubiquitin ligase controls spike development in wheat. Interestingly, this E3 ligase interacts with TaHIPP3 and promotes its degradation, suggesting a possible molecular mechanism involving HIPP proteins. The Arabidopsis HIPP3 homolog was reported to be involved in generative development by delaying the onset of flowering [67]. Interestingly, an HMA domain-containing non-prenylated protein, NaKR1/HPP2, has been shown to promote flowering by interacting with and regulating the long-distance movement of FLOWERING LOCUS T protein through sieve elements in Arabidopsis [100]. This suggests that HPP/HIPP proteins may play a broader role during flowering.

## 8. What Conclusions Are Currently Possible about PD-Associated HIPPs?

Here, we introduce HIPPs as a new group of PD-associated proteins and summarize the current knowledge that might help elucidate their role at PD. Eleven HIPPs and one HPP have so far been identified in PD proteomes, and for three of these, PD localization is experimentally confirmed (Table 1). However, so far no clearly PD-linked functions have been identified for any HIPP.

From the available data, no single functional theme emerges. For many of these proteins, no detailed functional studies have so far been conducted beyond the transcriptional association with various stresses. PD-associated isoforms are regulated by heavy metal, cold, drought, and hypoxia stresses [12,37,38,45,47,48,49,52], in response to virus infections and wound signals [41,46,53], by phosphatidylinositols [44] and by several plant hormones including cytokinin, auxin, brassinosteroids, and nitric oxide [39,40,55].

HIPP26, the most extensively characterized of these putative PD HIPPs, has been shown to be involved in cadmium detoxification [1,47] and drought responses [45,46,49] and to interact with a nuclear transcription factor [45], a PM-localized acyl-CoA-binding protein [47], a metallocarboxipeptidase inhibitor (tomato isoform) [51], and a PD-localized viral movement protein (*N. benthamiana* isoform) [46]. Its deletion results in no observable phenotype, but high redundancy has been observed amongst both clade I and clade IV HIPPs [1,33]. Thus, individual HIPP isoforms may have multiple functions, perhaps relating to their changing subcellular localizations or functional overlap with related family members, making it challenging to identify specific PD-associated roles.

The closest to a typical PD function is likely the increase in viral long-distance movement associated with the upregulation of drought responses by NbHIPP26, though PD permeability was not directly analyzed in this study [46]. On the other hand, functions of putative PD HIPPs, as far as they are known, do align with known roles of PD. The effects of heavy metals on PD have been summarized above, and there is evidence for functional relations of PD with drought and cold stress [101,102,103,104,105], whilst the literature on PD involvement in pathogen responses and plant development is large and beyond the scope of this review (see [18,87,88,89,106,107] for recent overviews). Thus, it is perhaps not surprising that PD-associated HIPPs might also participate in various different processes at PD. For example, it will be important to understand whether and how different ER-related processes, such as ERAD and ER stress, are functionally linked to PD, and how HIPPs control this.

A targeted analysis of what exactly the PD-specific functions of HIPPs are and how they fit into previously characterized signaling and stress response pathways has the potential to fill current gaps in knowledge about how PD perceive and process signals and integrate multiple vital aspects of terrestrial plant life.

## Data Availability

The review contains no unpublished data.

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
