# Peer review of "Heavy Metal-Associated Isoprenylated Plant Proteins (HIPPs) at Plasmodesmata: Exploring the Link between Localization and Function"

_plants, 2023, doi:10.3390/plants12163015_

Round 1

Reviewer 1 Report

Dear Authors,

The manuscript describe the current knowledge about the heavy metal-associated isoprenylated plant proteins (HIPPs) and they postulated role in the regulation of plasmodesmata is described. Manuscript describes in details the structure of these prpteins. The structure and role of plasmodesmata and the symplasmic communication is also described. The changes in symplasmic communication in the reaction to biotic and abiotic factors are presented. The manuscript present the knowledge about the current results from proteomic analyses indication the presence of these proteins in the plasmodesmata. Informations about the link between HIPPs and plant development in connection with the regulation of the plasmodesmata function are also provided.

                The topic of this manuscript is original in the scientific filed and summarizes the current knowledge in this field.

The literature is relevant and contains up-to-date articles, which is important for the reader, because it allows to get acquainted with the latest research results in this topic.

In the manuscript is one table and one figure which are well presented.

All my detailed comments are in the PDF file but comments are for text editing only. In two places in the text, instead of the number of references, there is the author's name; spaces are missing in several places; in species names, L. should not be written in italics.

Author Response

Dear Reviewer 1, thank you for your constructive review of our manuscript. We are uploading a detailed response here, as well as a revised version of the manuscript with tracked changes.

Reviewer 2 Report

The manuscript covers an important scientific data about a new group of proteins in vascular plants. I find the experoments well organized and the manuscriptwell written. I do reccomendthe manuscript tu further proceeding and publication in Plants.

English needs a minor editing.

Author Response

Dear Reviewer 2, thank you for your constructive review of our manuscript. We are uploading a response here, as well as a revised version of the manuscript with tracked changes.

Reviewer 3 Report

thank you for a fine review-  learned so much   

valued the interactions between roles in drought stress,  viral infections and heavy metal stresses  for the plasmodesmata -- intreguing how the plant uses its structures for all the stresses to which it may be exposed 

also the inclusion of what information is lacking is a feature not always included with weight  in reviews 

Author Response

Dear Reviewer 3, thank you for your constructive review of our manuscript. We are uploading a response here, as well as a revised version of the manuscript with tracked changes.